# Spectrofluorometric Analysis of Autofluorescing Components of Crude Serum from a Rat Liver Model of Ischemia and Reperfusion

**DOI:** 10.3390/molecules25061327

**Published:** 2020-03-14

**Authors:** Anna C. Croce, Andrea Ferrigno, Clarissa Berardo, Giovanni Bottiroli, Mariapia Vairetti, Laura G. Di Pasqua

**Affiliations:** 1Institute of Molecular Genetics, Italian National Research Council (CNR), Via Abbiategrasso 207, I-27100 Pavia, Italy; bottiroli@igm.cnr.it; 2Department of Biology & Biotechnology, University of Pavia, Via Ferrata 9, I-27100 Pavia, Italy; 3Department of Internal Medicine and Therapeutics, University of Pavia, Via Ferrata 9, I-27100 Pavia, Italy; andrea.ferrigno@unipv.it (A.F.); clarissa.berardo01@universitadipavia.it (C.B.); mariapia.vairetti@unipv.it (M.V.); lauragiuseppin.dipasqua01@universitadipavia.it (L.G.D.P.)

**Keywords:** liquid optical biopsy, liver injury, retinoids, vitamin A, free fatty acids, arachidonic acid

## Abstract

Autofluorescence (AF) of crude serum was investigated with reference to the potential of its intrinsic AF biomarkers for the noninvasive diagnosis of liver injury. Spectral parameters of pure compounds representing retinol (vitamin A) and fluorescing free fatty acids were characterized by spectrofluorometry, to assess spectral parameters for the subsequent AF analysis of serum, collected from rats undergoing liver ischemia/reperfusion (I/R). Differences in AF spectral profiles detected between control and I/R were due to the increase in the AF components representing fatty acids in I/R serum samples. No significant changes occurred for retinol levels, consistently with the literature reporting that constant retinol levels are commonly observed in the blood, except for malnutrition or chronic severe liver disease. Conversely, fatty acids, in particular arachidonic and linoleic acid and their derivatives, act as modulating agents in inflammation, representing both a protective and damaging response to stress stimuli. The biometabolic and pathophysiological meaning of serum components and the possibility of their direct detection by AF spectrofluorometry open up interesting perspectives for the development of AF serum analysis, as a direct, cost effective, supportive tool to assess liver injury and related systemic metabolic alterations, for applications in experimental biomedicine and foreseen translation to the clinics.

## 1. Introduction

The development of optical biopsy for multipurpose diagnostic applications in biomedicine relies on the presence of numerous types of biomolecules acting as endogenous fluorophores (EFs) [1,2,3]. In hepatology, multifaceted diagnostic perspectives are accounted for by the biochemical complexity of the liver and its engagement in numerous metabolic and catabolic functions to maintain the systemic homeostasis of nutrients and micronutrients, such as proteins, lipids, carbohydrates, vitamins, minerals, and electrolytes, and to detoxify the organism [4,5,6].

Attention has long since been given to NAD(P)H and flavins, as EF biomarkers of liver energy metabolism under normal, physiologically altered or pathological conditions [7,8,9]. Lipofuscins and lipofuscin-like lipopigments with their yellowish emission have accounted for the first observation of autofluorescence (AF) alteration in livers with metabolic disorder [10], and their intracellular accumulation as highly fluorescing granules is currently recognized as a marker of aging or of pathological conditions [11]. Collagen tends to accumulate under the stimulus of stressful events, stimulating the differentiation of stellate cells into myofibroblasts. Consequently, the bluish collagen AF, commonly found in connective tissue and extracellular matrix, is a suitable biomarker of inflammation and fibrosis [12]. Lipids are commonly not fluorescent. Only some free fatty acids can fluoresce when excited in the near-UV- blue spectral regions. A spectroscopic study on the AF of arachidonic acid as a pure compound in comparison with the AF of liver tissue extracts and living hepatocytes confirmed its effective contribution to the AF of liver tissue [13]. Since then, arachidonic acid has been considered to represent fluorescing lipids contributing to the overall AF of liver tissue. Changes in the contribution of fluorescing lipids to the overall liver tissue AF were also suggested to depend on the intrinsic metabolic properties of normal and fatty livers, rather than on functional alteration induced by external interferences, in a likely agreement with the metabolic plasticity of liver engaged in the maintenance of its own and whole organism homeostasis [14,15]. Subsequent work on pure compounds and lipid extract solutions from liver tissue showed the AF ability to discriminate arachidonic acid from other fluorescing free fatty acids, such as linoleic and oleic acids [16,17]. The semi quantitative analysis of the AF contribution of these lipid components, in parallel with mass spectrometry data, showed changes in the balance between arachidonic acid and a miscellanea of the other fluorescing free fatty acids in two different models of fatty liver, genetic or induced by a methyl-choline deficient (MCD) diet [18]. These studies generally investigated AF by using excitation light at 366 nm. This wavelength is particularly convenient, in fact it is compatible with the optical equipment of common microscopes and fiber optics, and it is obtained by using low-cost Hg lamps or 366 nm LEDs. Moreover, 366 nm wavelength is also convenient because it can excite simultaneously different EFs, allowing to collect EF signals by a single measurement and from a single site, maintaining their direct relationship with a metabolic activity in which they are involved, for example, in the case of optical ratio calculation. Artifacts from photobleaching are also minimized, avoiding the necessity to change the excitation conditions to repeat measurements on the same field [4,19].

Recently, preliminary data demonstrated the ability to detect fluorescing free fatty acids directly in native serum [20]. These findings and the possibility to easily change measuring conditions in liquid spectrofluorometric analysis offer new insights to improve serum biomarker panels and to advance noninvasive diagnostic applications in hepatology and, more generally, in the assessment of systemic metabolic alterations [21]. This suggestion is fully justified by the central role of liver in the accumulation, storage and mobilization of fatty acids for both metabolic supply and signaling regulation of harmful or protective response to hazardous stimuli in the intrahepatic and extrahepatic progression of diseases [22,23,24,25,26].

Liver is also a target organ for the regulatory actions of vitamin A and retinoids, also contributing to their storage and maintaining constant their healthy levels in the blood [6,27,28]. Vitamin A is indispensable for many body functions, including liver glucose and lipid metabolism, cell proliferation and differentiation, immune response, development and function of nervous system, and reproduction and vision [6,28]. As a consequence, researchers have long since been focusing on defining reference blood levels of vitamin A or, more generally, retinoids, and on the development of reliable assay procedures, traditionally to assess vitamin A deficiency [29,30]. An initial application based on a direct fluorometric measurement of the intrinsic AF of retinol was followed by the development of more complex procedures based on liquid or gas chromatography, mass spectroscopy, and high performance liquid chromatography, improving sensitivity and specificity of analysis procedures [29,31,32]. Fluorometric measurement of retinol in blood remains, however, a valuable possibility for a rapid and cost effective detection of vitamin A, simultaneously with additional EFs [4,30,33].

In this work, we have first characterized in more depth the AF spectrofluorometric properties of retinol and arachidonic acid along with oleic and linoleic acids, representing additional fluorescing free fatty acids. Various excitation and emission conditions have been considered, including the 366 nm excitation, which has been already extensively used in our AF studies on liver tissue [4]. The defined spectral parameters were then applied to the analysis of the AF of serum collected from rats undergoing liver ischemia and reperfusion [34]. Our primary objective has been to validate a procedure to analyze crude serum AF and estimate the contribution of AF spectral components, representing different EFs relatable to liver functionality. The changes in the AF components in livers responding to external stress stimuli are a promising supportive diagnostic tool for both intrahepatic injury and alteration of the organ engagement in the maintenance of systemic homeostasis.

## 2. Results

### 2.1. Spectrofluorometric Analysis of Pure Compounds

Pure compounds representing retinol and free fatty acids showed different spectral positions with different extents of changes depending on the wavelength used to record or excite the AF signal. Retinol exhibited excitation and emission spectra centered at about 320 nm and 490 nm, independently from the respective reading or excitation wavelength (Figure 1A).

When compared with retinol, oleic acid showed excitation and emission spectra respectively at longer and shorter wavelengths. The position of excitation spectra centered at about 350 nm was still independent from the wavelength of observation, while emission shifted to longer wavelengths when increasing excitation. The peak position ranged from about 415 nm to about 430 nm under the respective excitation at 310 nm and 366 nm (Figure 1B).

Linoleic acid spectra showed the most remarkable variability. The peak of the excitation spectrum ranged from 300 nm to 360 nm, when observed at 380 nm and 510 nm. Interestingly, excitation spectra recorded at 440 and 460 nm showed a peak respectively at 320 and 340 nm and a shoulder at about 360 nm, while spectra recorded at 510 nm showed a peak at about 360 nm and a shoulder at about 320 nm. These findings suggest the presence of two bands centered at about 320 and 360 nm, the prevalence of which depended on the reading wavelength. Emission spectra, in turn, showed a remarkable change in the peak position, increasing by about 100 nm under 366 nm as compared with 310 nm excitation. Notably, the maximum peak observed at about 460 nm under 366 nm excitation corresponded to the position of the weak shoulder of the spectrum obtained under 310 nm excitation (Figure 1C). Lastly, arachidonic acid excitation spectra did not evidence remarkable changes in the main band, peaking at about 315 nm. Increasing the reading wavelength resulted in the narrowing of the main excitation band and in the rising in the relative amplitude of the shoulder observable in the 350–370 nm interval. Conversely, the main emission band showed a remarkable shift toward longer wavelength, when observed under the respective excitations at 310 nm to 340 nm and 366 nm, with maximum peak positions at about 420 nm, 450 nm and 470 nm (Figure 1D).

The direct comparison of the excitation profiles recorded from the different fluorophores made it easy to appreciate the effect of the reading wavelengths at a shorter or longer position than the emission peak of retinol, taken as a reference because of its fixed position. Excitation peaks of retinol and arachidonic resulted to be closer to 310 nm than to 366 nm, regardless of the reading wavelength. The wide excitation spectral shape of oleic and linoleic acids covered both 310 and 366 nm positions. However, the conversion of the two bands in the linoleic acid excitation profile made the 366 nm excitation more favorable for its detection when the signal was measured at 510 nm (Figure 2A,B).

The grouping of the emission profiles, in turn, made it easier to appreciate that arachidonic and linoleic acid spectra were overlaying retinol emission to a higher degree when excited at 366 nm than at 310 nm (Figure 3A,B).

The emission profiles recorded from pure compounds under 310 and 366 nm excitation were then submitted to spectral fitting analysis, to define the parameters to be used for the AF emission spectra collected from rat serum.

### 2.2. Liver Injury Biochemical Assays

The common biomarkers of liver injury were first assayed biochemically to confirm the level of ischemia reperfusion (I/R) injury. Table 1 summarizes data on aspartate transaminase (AST), alanine transaminase (ALT), alkaline phosphatase (ALP) assayed in the serum, and on inducible Nitric Oxide Synthase (iNOS) assayed in the liver tissue [35,36].

### 2.3. Spectrofluorometric Analysis of Serum

Spectrofluormetric analysis of AF emission from serum samples showed spectra consisting in a wide band in the 410–590 nm emission range, under excitation at either 310 nm o 366 nm (Figure 4A,B).

Commonly, spectra showed a maximum peak position around 450 nm, with an emission tail at the shorter wavelength side of the spectrum, consistent with a contribution of serum proteins, mostly albumin. Spectra showed also a shoulder in the 490–510 nm region, which decreased after I/R, as compared with sham-operated rats. The shoulder and its changes were more appreciable under excitation at 310 nm than at 366 nm.

The contribution to the overall AF signal of the spectral components representing the different EFs was estimated by curve fitting analysis by using the spectral parameters defined for each EF possibly present in serum (Figure 5A–F). The power of fitting analysis allows a numeric estimation of small changes in spectral shape [8]. In this case, the achievement of the goodness of fitting required the addition of minor bands, one ascribable to the undefined contributor, and two likely due to the negative effect of hemoglobin or to flavins. The results of fitting analysis are summarized in Table 2. AF data relatable to linoleic acid for 310 nm excitation, AF data relatable to proteins for 366 nm excitation, and AF data of minor bands are not given because of values ≤5%.

The excitation at 310 nm gave rise to spectra mostly accounted for by the AF of retinol and arachidonic acid, with a lesser contribution by proteins and oleic acid, and minor values ascribable to linoleic acid. Spectra recorded under 366 nm excitation were still mostly accounted by the retinol and arachidonic acid AF, but with a remarkable increase in the contribution of AF relatable to linoleic and oleic acids, and much lesser values for proteins. The serum samples from rats undergoing liver I/R in comparison with sham-operated showed a decrease of the relative AF contribution of retinol and an increase in the AF of arachidonic acid under both 310 nm and 366 nm excitations (Table 2). The 366 nm excitation also allowed to estimate an increase in the AF contributions relatable to linoleic and oleic acids in I/R, as compared with sham-operated rats. The estimation of the relative contributions of the various AF components thus allowed to assess their balance and compare changes between sham-operated and I/R serum samples.

Additional valuable information was provided by the estimation of the real, AF measured spectral data. The integrated values of the measured AF spectral areas were much higher under 310 nm than under 366 nm. Under both excitation conditions, signals were significantly higher after IR 60/60 min (Figure 6).

Besides, the real AF contribution of each AF component estimated from the percentage values on the basis of the integrals of the measured AF areas allowed to compare the effective incidence of the EFs between sham-operated and I/R samples.

Data summarized in Table 3 indicate that, depending on the overall AF area values being much higher in I/R samples than in controls, the differences in the AF contribution ascribable to retinol are greatly diminished, while the differences in the arachidonic acid AF values are significant under both 310 nm and 366 nm. This condition resulted also in significantly higher values for both linoleic and oleic acids AF contribution in I/R serum samples as compared with sham-operated rats.

## 3. Discussion

The preliminary investigation of the AF spectral properties of serum EFs, performed on pure compounds representing retinol and FFFAs in ethanol solution, showed a variability in their spectral positions depending on the wavelength applied for the analysis. The greatest changes in both excitation and emission spectra were seen with linoleic acid, while retinol did not undergo any remarkable change. These events may be explained according to the chemical structure of these compounds and a possible phenomenon of energy delocalization (Figure 7, in Materials and Method). The number and position of the double bonds along the linear carbon chain can influence the internal redistribution of excitation energy, affecting the energy levels with consequent changes in the wavelength position of the emitted light [37]. The greatest dependence of spectral position on the interrogating wavelength was exhibited by linoleic acid, in particular as to the increasing position of the emission when excitation wavelength was increased. These findings are consistent with the two conjugated double bonds of linoleic acid, a condition favorable to split the energy of the two orbitals, resulting in different levels of energy in the excited state. Conversely, in retinol the presence of methyl groups along the linear carbon chain and of a cyclohexene ring is likely preventing an uneven distribution of the energy in the excited state, accounting for the fixed position of its excitation and emission spectra. As a consequence, the photophysical properties of retinol and FFFAs may explain the greater ability of the 310 nm excitation to discriminate the different compounds as compared with 366 nm, which resulted in the red shift of the emission of arachidonic and linoleic acids toward that of retinol.

As for serum AF spectra, both 310 nm and 366 nm excitation resulted in a wide emission band in the 420–520 nm interval. The excitation at 310 nm also evidenced a shoulder in the 480–520 nm region, which was more marked in sham-operated than in I/R serum samples. The shoulder was less evident under excitation at 366 nm, resulting in a common widening of AF spectra towards longer wavelengths.

The changes in serum AF spectral shape reflected changes in the contribution of the various EFs. Curve fitting analysis showed retinol and arachidonic acid as the main AF components of the overall spectrum recorded under 310 nm excitation. Retinol AF contribution was lower in I/R samples than in sham-operated ones, compensated by the relative increase in the arachidonic acid AF. Comparable results were obtained from spectra recorded under 366 nm excitation, thanks to the ability of spectral fitting analysis to discriminate the AF contribution relatable to arachidonic acid regardless of the red shift occurring under 366 nm excitation. With respect to 310 nm, the 366 nm excitation resulted also in more suitable spectral conditions to detect the AF relatable to linoleic and oleic acids, allowing to find an increase in their AF values in I/R as compared with sham-operated rat samples.

According to the profiles of the excitation spectra of the EFs, the integrated values of the overall measured spectral areas were higher for 310 nm than 366 nm excitation, and for both excitation conditions the overall AF areas were significantly higher in I/R than in sham-operated rats. The use of the overall integrated AF values to correct the relative AF spectral contribution of each EF provided information on the real contribution of the AF components in the different serum samples. Differently from the AF percentage data, the real AF values of retinol showed no significant difference between sham-operated and I/R 60/60 and IR 60/120 min serum samples. On the other hand, the real AF contribution of arachidonic acid remained higher in I/R 60/60 and IR 60/120 min than in control samples under both 310 and 366 nm excitation. The 366 nm excitation showed also that the real AF values of linoleic and oleic acids were significantly higher in I/R 60/60 and IR 60/120 min samples than in sham-operated rats.

Our findings on AF relatable to retinol indicate that its levels in the blood were not greatly affected by our experimental conditions applied to induce liver I/R. This finding is consistent with the literature, reporting that serum vitamin A is normally affected only by malnutrition or severe, chronic liver disease. When vitamin A supplementation is needed, it is important to maintain the balance between its antioxidant and toxic effects. In fact, in diseased liver, owing to the decreased ability of liver to produce RBP to export vitamin A in serum, vitamin A accumulation may occur, resulting in hepatic toxicity [38]. In this respect, we may recall that AF of retinol is strongly enhanced by binding to RBP and that the correlation between their concentrations has been proposed to estimate both components by the fluorometric measure of only one of them [32,33].

Similarly to vitamin A, the standard serum levels in healthy individuals for cholesterol, triglycerides, low and high density lipoproteins (LDL, HDL), and total free fatty acids, to be used as reference values for diagnostic purposes, have long since been established. The necessity to establish standard reference levels also for single classes of free fatty acids has been stressed by Abdelmagid et al. [39], addressing the need to help advances in the understanding of pathophysiological mechanisms involving these compounds and to improve the application of diagnostic data in the biomedical field. Currently, work is ongoing on free fatty acids and their many derivatives acting as signaling molecules in inflammatory pathways [22,23]

In this direction, the profiling of free arachidonic and linoleic acids in plasma and of their several derivatives, in particular eicosanoid derived from arachidonic acid, has been proposed to improve the panels of non-invasive biomarkers of fatty liver and disease progression [40]. Attention has also been given to the induction of hepatic stress by events such as I/R with the consequent activation of oxidative stress and of the cascade production of eicosanoid derivatives of arachidonic acid, mediators of inflammation and microcirculation alteration, eventually causing liver injury [26]. With regards to our results, we may underline that our AF data indicating an increase in serum free fatty acids in rats undergoing liver I/R are in agreement with a report on a mouse model for which liver I/R resulted in blood enrichment in microparticles containing increasing levels of fatty acids and their derivatives [41].

## 4. Materials and Methods

### 4.1. Pure Compounds and AF Spectrofluorometric and Fitting Analysis

Fluorescence of pure compounds in ethanol 100% solution (Figure 7) was analyzed by spectrofluorometry (Spectrofluorometer LS 55 Model; PerkinElmer Italia, Milan, Italy).

For each pure compound, a preliminary set of measurements was performed under variable conditions of excitation and emission conditions, to detect the excitation and emission positions of major peaks, and to assess the most suitable concentrations to collect the spectra (retinol 1 × 10^−6^ M; arachidonic acid 1 × 10^−3^ M, oleic acid 1 × 10^−1^ M, linoleic acid 1 × 10^−2^ M). The analyses addressed the choice of the excitation wavelengths at 310 nm and 366 nm, the latter matching with excitation wavelength previously applied by us to investigate AF of liver tissue, ex vivo and in vivo [4,42].

The AF emission spectra of pure compounds were used to define the starting parameters describing the Half-Gaussian Modified Gaussian (GMG) spectral functions, in terms of wavelength position of the central peak (λ) and full width at the half intensity maximum (FWHM) as respectively reported for excitation at 310 nm: retinol (490 nm, 112 nm), arachidonic acid (425 nm, 120 nm), oleic acid (370 nm, 85 nm), linoleic acid (417 nm, 92 nm), and for excitation at 366 nm: retinol (490 nm, 112 nm), arachidonic acid (470, nm, 93 nm), oleic acid (462 nm, 90 nm), and linoleic acid (428 nm, 73 nm). The component representing the emission tail of proteins at wavelengths <420 nm was made free to adapt for the goodness of fitting. The goodness of fitting required the addition of minor bands. The band peaking at about 440 nm matches with emission regions of fluorescing fatty acids and of NAD(P)H, but it is indicated as undefined, being at this time difficult to assign it to a defined fluorophore. Conversely, the negative band peaking at about 410 nm might be due to hemolysis and not completely avoidable during the blood processing to obtain serum. The consequent interference of hemoglobin absorption was likely the cause of the distortion of the remarkable emission signal observed in the region around 410 nm when spectra were excited at 310 nm, to be compensated with the negative band. The band peaking at about 560 nm, observed only in I/R samples analyzed under 366 nm excitation, may indicate an increase in flavins released by injured livers. This suggestion is consistent with a recent report on the increase in mitochondrial flavins detected in the perfusate of liver submitted to hypothermic oxygenated perfusion for transplantation [43] and with the spectral unmixing approach based on detection of three bands in flavin fluorescence, one of which peaks at 560 nm [44].

Spectral parameters were used for the subsequent fitting analysis of serum AF spectra. Ethanol and serum may differently affect the fluorophore spectral position and shape. To overcome the problem, a satisfying goodness of fitting is first achieved on the control by using starting spectral parameters, and the so defined control GMG combination are then used as a starting point to proceed to the comparative analysis of I/R samples. Before analysis, spectral peaks were also normalized to 100 a.u. to avoid extra and unnecessary calculations to adapt to amplitude changes searching only for shape changes, that is changes in the contribution of each single AF component to the overall emission area. Hence, each single AF component are expressed as a percentage of the overall area [13], and absolute data are then obtained from the percentages corrected for the real measured amplitude values. The data processing is based on the iterative non-linear curve-fitting procedure (PeakFit; SPSS Science, Chicago, IL), using the Marquardt-Levenberg algorithm [45]. The goodness of fitting is verified by the definition of the true absolute minimum value of the sum of squared deviations and by the analysis of residuals and the coefficient of determination (r^2^).

### 4.2. Animal Models

Male Wistar rats (220–250 g, *n* = 21, total; Harlan-Nossan, Correzzana, Italy) were maintained in 12 h of dark/light cycles, controlled temperature (21 °C), and given free access to water and food.

The effects of I/R were studied in vivo in a partial normothermic hepatic I/R model. Liver ischemia and subsequent reperfusion were induced in rats anesthetized with in pentobarbital (40 mg/kg). The abdomen was opened via a midline incision, and ischemia was induced to the left and the median lobe for 60 min with microvascular clips by clamping the branch of the portal vein and the branch of the hepatic artery after the bifurcation to the right lobe. The abdomen was temporarily closed with a suture. After 60 min of ischemia, the abdomen was reopened, the clips were removed, the abdomen was closed again, and the liver was allowed to reperfuse for 60 or 120 min. By using partial, rather than total, hepatic ischemia, portal vein congestion and subsequent bacterial translocation into the portal venous blood was avoided [46]. Sham animals were subjected to the same procedure without clamping the vessels (*n* = 7). To prevent postsurgical dehydration and hypotension, 1 mL of saline was injected into the inferior vena cava. All the animals were maintained on a warm support to prevent heat loss (rectal temperature at 37 ± 0.1 °C). Blood was drawn from the vena cava and allowed to clot at room temperature. After 15 min, blood samples were centrifuged to separate serum, which was frozen in liquid N_2_ and stored at −80 °C until being processed for AF spectroscopic analysis. Hepatic biopsies were quickly removed from the median lobe and immediately frozen in liquid nitrogen for subsequent biochemical assays. The use and care of animals in this experimental study was approved by the Italian Ministry of Health and by the University of Pavia Commission for Animal Care (Document number 179/2017-PR).

Liver injury was assessed by serum level evaluation of alanine transaminase (ALT), aspartate transaminase (AST) and alkaline phosphatase (AP) using commercial kits (Sigma).

A Western Blot assay was used to evaluate iNOS from liver tissue samples by using rabbit polyclonal antibody against iNOS (Cayman Chemical, Ann Arbor, Michigan, USA) following a procedure already described in detail [47].

### 4.3. Statistical Analysis

Unistat (Unistat^®^ Statistical Package, Version 6.5 04, Unistat Ltd., London, England) and R software (R Development Core Team) were used to perform statistical analysis. The value of *p* <0.05 was considered as statistically significant.

## 5. Conclusions

Spectral fitting analysis of AF of crude serum allowed to directly detect changes between I/R and sham-operated rats, reflecting changes in the balance of fluorescing components ascribable to retinol and fluorescing free fatty acids. These changes were found to depend mostly on the real increase in the AF contribution relatable to fluorescing free fatty acids rather than to variations in the AF relatable to retinol. Our findings are in line with the already recalled reports in the literature on the role of hepatic stress in inducing a cascade release of arachidonic acid and production of eicosanoid derivatives in the sensing and regulation of inflammation and microcirculation as causes of liver injury [26]. Furthermore, our results are supported by the blood enrichment in microparticles with increasing content in fatty acids and their derivatives reported in a mouse model of I/R [41].

Although not strictly quantitative, as biochemistry could do, our results on the AF analysis of serum open promising perspectives to improve the serum panels of biomarkers to advance the non-invasive, real time and cost-effective estimation of serum fluorescing compounds as significant biomarkers of pathophysiological pathways in the progression of liver injury. The AF properties of serum are thus worthy of further investigation on additional models of liver disease, such as fatty livers and oxidative stress, to advance the application of AF analysis as supportive diagnostic tools in hepatology, for applications in experimental biomedicine for the assessment and monitoring of liver metabolic functionality, and its enrollment in the management of metabolic systemic homeostasis, with a foreseen translation to the clinics.

## Figures and Tables

**Figure 1 molecules-25-01327-f001:**
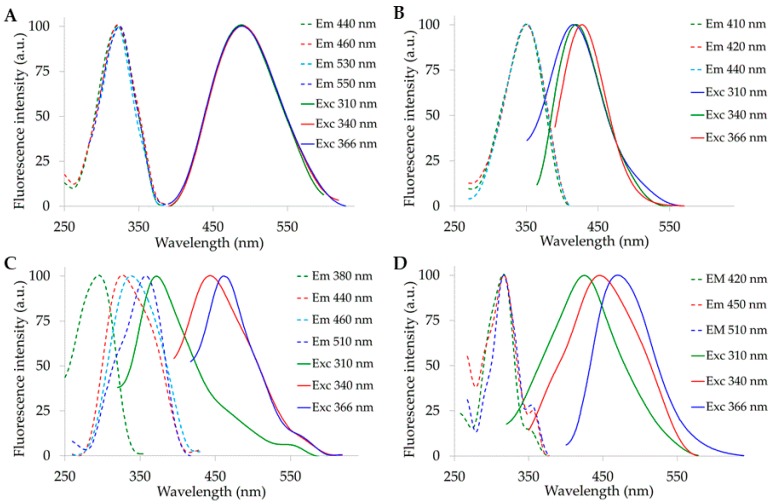
Fluorescence excitation and emission spectra of pure compounds in ethanol solution (100%). (**A**) retinol 1 × 10^−6^ M; (**B**) oleic acid 1 × 10^−1^ M; (**C**) linoleic acid 1 × 10^−2^ M; (**D**) arachidonic acid 1 × 10^−3^ M. Spectra are normalized to the maximum peak value. Excitation and reading emission wavelengths are indicated by colors, on the right.

**Figure 2 molecules-25-01327-f002:**
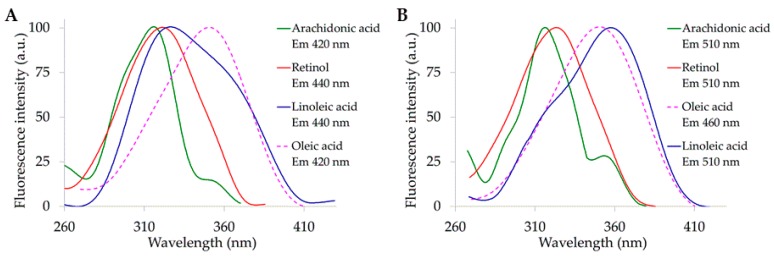
Excitation spectra of pure compounds in ethanol solution (100%), observed at wavelength shorter (**A**) or longer then (**B**) 450 nm. Concentrations are reported in the Figure 1 caption. Spectra are normalized to the maximum peak value. Compounds and reading emission wavelengths are indicated by colors on the right.

**Figure 3 molecules-25-01327-f003:**
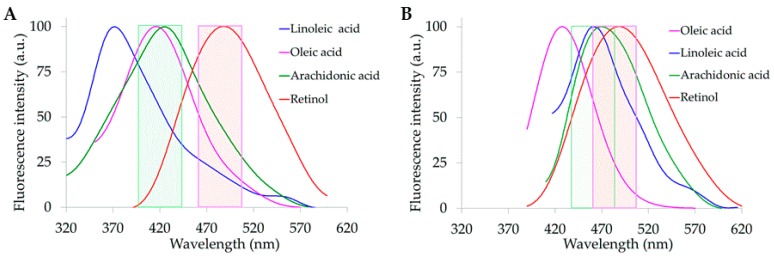
Emission spectra of pure compounds in ethanol solution (100%), excited at 310 nm (**A**) or 366 nm (**B**). Concentrations are reported in the Figure 1 caption. Spectra are normalized to the maximum peak value. Compounds are indicated by colors, on the right.

**Figure 4 molecules-25-01327-f004:**
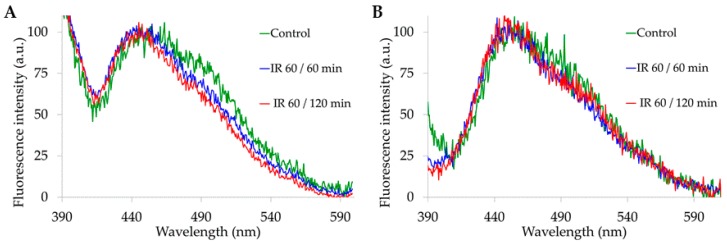
Examples of emission spectra recorded from serum samples under (**A**) 310 nm and (**B**) 366 nm excitation. Spectra are normalized to the maximum peak value. Rat liver treatment conditions are indicated by colors.

**Figure 5 molecules-25-01327-f005:**
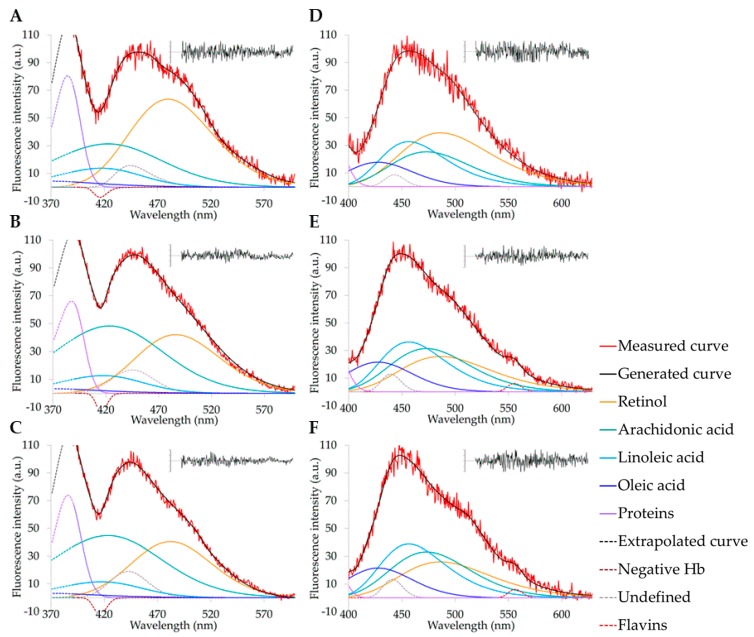
Examples of curve fitting analysis of spectra recorded under (**A**–**C**) 310 nm and (**D**–**F**) 366 nm excitation from serum of sham operated (**A**,**D**), I/R 60/60 min (**B**,**E**) and I/R 60/120 min (**C**,**F**). Spectra are normalized to the maximum peak value. AF spectral components are defined by colors. Dashed lines represent extrapolated part of curves from Peak fit program (310 nm excitation, left side of spectra recorded in the 390–630 nm interval) or undefined components. Goodness of fitting was verified by analysis of residuals (inset graphs) and by coefficient of correlation (r^2^ ≥ 0.95).

**Figure 6 molecules-25-01327-f006:**
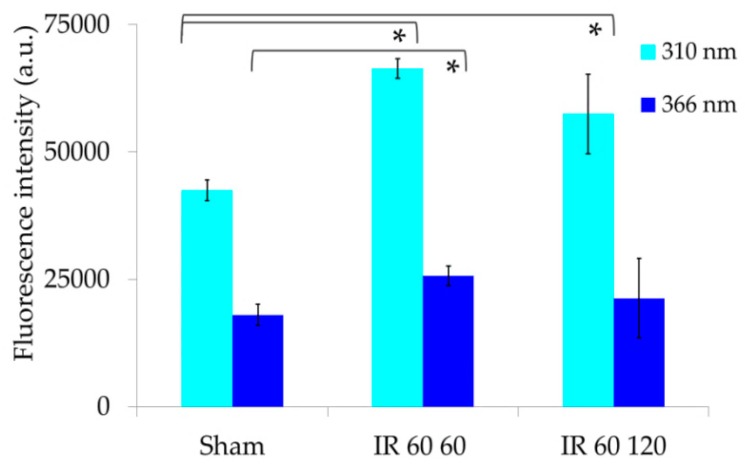
Bar chart representing the integrated values (Means ± S.E.) of the overall emission areas calculated from the AF spectra recorded from serum samples under respective 310 nm and 366 nm excitation, **p* ≤ 0.05.

**Figure 7 molecules-25-01327-f007:**
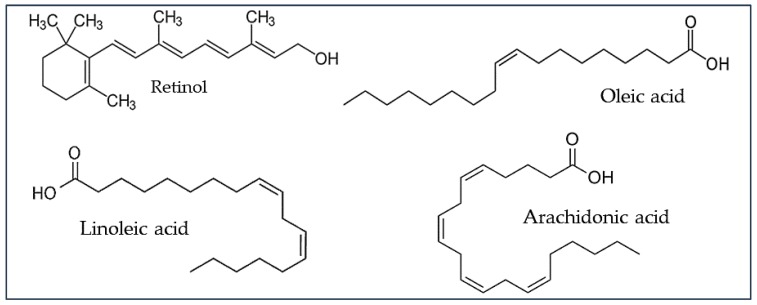
Structure formulae and pure compounds.

**Table 1 molecules-25-01327-t001:** Biomarkers of liver injury, serum levels of AST, ALT and ALP, and tissue levels of iNOS, (Means ± S.E.).

		Control	I/R 60 *	I/R 120 *
**Serum**				
	AST	254.63 ± 59.12	3503.23 ± 1049.05	10405.36 ± 1099.88
	ALT	69.17 ± 22.91	3911.07 ± 893.58	9403.55 ± 1055.46
	ALP	428.33 ± 20.47	613.28 ± 58.91	772.05 ± 31.27
**Liver Tissue**				
	iNOS	0.89 ± 0.11	1.53 ± 0.21	2.26 ± 0.26 *

Statistical analysis: data in I/R 60/60, I/R 60/120 columns versus respective controls: * *p* ≤ 0.05.

**Table 2 molecules-25-01327-t002:** Relative contribution (%) of the different endogenous fluorophores to serum AF spectral area, estimated by curve fitting analysis (Means ± S.E.).

Liver Treatment Time	Endogenous Fluorophores					
	Retinol (Ret)	Arachidonic Acid (AA)	Linoleic Acid	Oleic Acid	Proteins	(AA)/(Ret)
**310 nm excitation**						
*Sham-operated*	44.26 ± 1.95	25.13 ± 1.12	----	7.93 ± 0.39	12.48 ± 0.60	0.58
*IR 60/60 min*	31.39 ± 0.84 **	38.93 ± 1.04 **	----	8.86 ± 0.23	8.20 ± 0.21	1.23
*IR 60/120 min*	29.09 ± 3.63 **	39.16 ± 4.95 *	----	8.56 ± 0.90	10.24 ± 1.45	1.36
**366 nm excitation**						
*Sham-operated*	33.43 ± 2.68	20.22 ± 1.52	23.32 ± 1.86	12.88 ± 1.03	----	0.60
*IR 60/60 min*	24.03 ± 1.35 *	26.50 ± 1.49 *	27.47 ± 1.55	16.97 ± 0.67 *	----	1.10
*IR 60/120 min*	23.76 ± 3.34 *	27.17 ± 3.82	27.02 ± 3.93	16.97 ± 2.38	----	1.14

Statistical analysis I/R 60, I/R 120 versus sham: **p* ≤ 0.05; ** *p* ≤ 0.01.

**Table 3 molecules-25-01327-t003:** Real contribution of the different endogenous fluorophores to the serum overall AF spectral area (390–600 nm range). (Means ± S.E.).

Liver Treatment Time	Endogenous Fluorophores				
	Retinol (Ret)	Arachidonic Acid (AA)	Linoleic Acid	Oleic Acid	Proteins
**310 nm Excitation**					
*Sham-* *operated*	18786.35 ± 828.23	10826.07± 477.01	----	3791.60 ± 167.59	5880.18 ± 251.68
*IR 60/60 min*	20694.20 ± 559.98	25811.30 ± 688.32**	----	5883.34 ± 157.96	5447.30 ± 145.07
*IR 60/120 min*	16680.27 ± 2080.75	22768.95 ± 2848.29**	----	4350.10 ± 545.00	6654.47 ± 833.74
**366 nm Excitation**					
*Sham-* *operated*	6034.63 ± 438.46	3648.04 ± 292.31	4208.95 ± 337.17	2326.57 ± 186.43	----
*IR 60/60 min*	6180.46 ± 349.19	6809.04 ± 384.22 **	7077.63 ± 399.72 **	4377.84 ± 246.91 *	----
*IR 60/120 min*	5056.64 ± 710.25	5780.31 ± 812.19 *	5953.46 ± 836.49 *	3611.97 ± 507.93*	----

Statistical analysis I/R 60, I/R 120 versus sham: * *p* ≤ 0.05; ** *p* ≤ 0.01.

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
