# Peer review of "Spectrofluorometric Analysis of Autofluorescing Components of Crude Serum from a Rat Liver Model of Ischemia and Reperfusion"

_molecules, 2020, doi:10.3390/molecules25061327_

Round 1

Reviewer 1 Report

The present manuscript has in attention as main objective the  validation of a procedure of analyzing crude serum AF and estimate the contribution of autoflourescent spectral components, representing different endogeneous fluorophores regarding the liver functionality. 

The manuscript is interesting, but it needs minor English revisions and/or spell checking. 

The values in the tables, as well as in the text should have the same number of decimals. 

A short comment regarding the presence of the undefined components should be considered for inclusion in the manuscript.

For the readers convenience, the authors are kindly requested to revise the paragraph from line 216-220, as the explanations are difficult to be followed.

„Conclusions” should be reconsidered; it is commendable to refer to the obtained results and make a brief comparison with the existing published similar results.

Author Response

Reviewer 1

We want to thank the Reviewer for the meaningful comments, which surely helped us to improve the paper. English and text revision according to the comments are evidenced in yellow. No evidences are given in Discussion, since the overall text, although maintain the sequence of arguments unchanged, was entirely revised.

The present manuscript has in attention as main objective the validation of a procedure of analyzing crude serum AF and estimate the contribution of autoflourescent spectral components, representing different endogeneous fluorophores regarding the liver functionality.

The manuscript is interesting, but it needs minor English revisions and/or spell checking.

Q1 – The values in the tables, as well as in the text should have the same number of decimals.

A1 – The text has been checked. Decimals have been added to values in Table 1, and one correction has been made in Table 2.

Q2 - A short comment regarding the presence of the undefined components should be considered for inclusion in the manuscript.

A- 2 Some few comments on undefined components, namely negative band at 410 nm, were already given in M&M (old text, lines 314-317). Now the text has been revised by adding new information (lines 180-184; lines 327-339). Figure 5 has been also updated.

Q -3 For the readers convenience, the authors are kindly requested to revise the paragraph from line 216-220, as the explanations are difficult to be followed.

A- 3 With agree with the Reviewer that the text was confused, and provided to revise it.

Q – 4 “Conclusions” should be reconsidered; it is commendable to refer to the obtained results and make a brief comparison with the existing published similar results.

A – 4 We fully agree with this comment, and we have revised “Conclusions” text.

Reviewer 2 Report

The manuscript by A.C. Croce et al. reports on measurements of intrinsic fluorescence of crude serum upon liver injury (by ischemia and reperfusion) in comparison with control measurements. In addition, individual constituents (retinol, fluorescent fatty acids) are examined in ethanol solution. A main result is that the contributions of individual components to integral fluorescence, in particular arachidonic acid and oleic acid, change considerably upon injury, whereas retinol remains almost constant. Therefore, analysis of the contributions of these fluorophores may support diagnosis of liver injury and related metabolic alterations.

The manuscript is well and clearly written, and I have only a few comments which should be addressed in a revised version:

  1. I suppose that the caption of Fig. 3 is wrong. Probably this should be emission spectra after excitation at 310 nm and 366 nm, respectively.
  2. Changes due to liver injury, as depicted in Fig. 4, are rather small. Therefore the pronounced difference of the contributions of individual fluorophores, as  reported in the Tables 2 and 3, is rather astonishing. Perhaps the authors could comment on this. I am also missing a comment on whether the spectra of individual fluorophores (at the excitation wavelengths of 310 nm and 366 nm) are expected to be similar in ethanol and in serum.
  3. In addition to retinol, fatty acids and proteins there is some undefined fluorescence in Fig. 5. Do the authors have any idea about this fluorescence signal? Could they possibly correlate it with NADH or flavins?

Author Response

Reviewer 2

We want to thank the Reviewer for the meaningful comments, which surely helped us to improve the paper. English and text revision according to the comments are evidenced in yellow. No evidences are given in Discussion, since the overall text, although maintain the sequence of arguments unchanged, was entirely revised.

The manuscript by A.C. Croce et al. reports on measurements of intrinsic fluorescence of crude serum upon liver injury (by ischemia and reperfusion) in comparison with control measurements. In addition, individual constituents (retinol, fluorescent fatty acids) are examined in ethanol solution. A main result is that the contributions of individual components to integral fluorescence, in particular arachidonic acid and oleic acid, change considerably upon injury, whereas retinol remains almost constant. Therefore, analysis of the contributions of these fluorophores may support diagnosis of liver injury and related metabolic alterations.

The manuscript is well and clearly written, and I have only a few comments which should be addressed in a revised version:

Q 1 - I suppose that the caption of Fig. 3 is wrong. Probably this should be emission spectra after excitation at 310 nm and 366 nm, respectively.

A 1 – The Reviewer is right, and we apologize for the mistake. The correct legend is now given for Figure 3.

Q 2.1 - Changes due to liver injury, as depicted in Fig. 4, are rather small. Therefore the pronounced difference of the contributions of individual fluorophores, as reported in the Tables 2 and 3, is rather astonishing. Perhaps the authors could comment on this.

A 2.1 – I agree with the Reviewer that differences in spectra seem small. However, the power of fitting analysis was already helping to discriminate small spectral differences. A sentence with a related reference (Croce et al., 2017) has been added to the text (lines 179-180).

Anyway, it is to consider that the values given as percentages do not show great differences for very similar spectra. For example, in the case of 310 nm excited spectra, major differences occur between sham and I/R, and small or almost no differences occur between I/R 60/60 and 60/120 min. The same for 366 nm excitation. As to Table 3, the changes in AF intensity (Figure 6) influence the real data of fluorophores, but the greatest differences are still between sham and I/R (60/60, and 60/120).

Q 2.2 – I am also missing a comment on whether the spectra of individual fluorophores (at the excitation wavelengths of 310 nm and 366 nm) are expected to be similar in ethanol and in serum.

A 2.2 – We agree with the Reviewer on this comment. It is commonly known that the environment can affect the spectral shape and position. Aware of the problem, in former work (Croce et al., 2004) we provided to compare pure compounds and cell extracts in same solvents, to verify and validate acceptable reference spectra to be then used to analyze “biological samples”. So we could take advantage of autofluorescence and spectral fitting analysis to get information on the biometabolic and functional state of our “biological substrates”, i.e. cells, liver tissue, bile, and so on. Besides, since analyses are made on series of the same “substrate” (for example liver tissue) coming from normal (control) or altered or diseased organ, we always start with the “control” condition. When the fitting is satisfying, we fix the combination of spectra and start from this to go to the processing of samples obtained under altered or diseased conditions. Text has been revised accordingly (lines 340-350).

Q 3 –In addition to retinol, fatty acids and proteins there is some undefined fluorescence in Fig. 5.

Do the authors have any idea about this fluorescence signal? Could they possibly correlate it with NADH or flavins?

A 3- Some few comments on undefined components, namely negative band at 410 nm, were already given in M&M (old text, lines 314-317). Now the text has been revised by adding new information (lines 180-184; lines 327-338). Figure 5 has been also updated. In this respect, we want to thank the Reviewer in helping us to improve the paper with considerations on flavins.